# Stem cell regionalization during olfactory bulb neurogenesis depends on regulatory interactions between *Vax1* and *Pax6*

Nathalie Coré[1]*, Andrea Erni[1], Hanne M Hoffmann[2], Pamela L Mellon[2], Andrew J Saurin[1], Christophe Beclin[1], Harold Cremer[1]*

[1]Aix Marseille Univ, CNRS, IBDM, Campus de Luminy, Marseille, France; [2]Department of Obstetrics, Gynecology, and Reproductive Sciences and the Center for Reproductive Science and Medicine, University of California, San Diego, San Diego, United States

**Abstract** Different subtypes of interneurons, destined for the olfactory bulb, are continuously generated by neural stem cells located in the ventricular and subventricular zones along the lateral forebrain ventricles of mice. Neuronal identity in the olfactory bulb depends on the existence of defined microdomains of pre-determined neural stem cells along the ventricle walls. The molecular mechanisms underlying positional identity of these neural stem cells are poorly understood. Here, we show that the transcription factor Vax1 controls the production of two specific neuronal subtypes. First, it is directly necessary to generate Calbindin expressing interneurons from ventro-lateral progenitors. Second, it represses the generation of dopaminergic neurons by dorsolateral progenitors through inhibition of Pax6 expression. We present data indicating that this repression occurs, at least in part, via activation of microRNA miR-7.

**\*For correspondence:**
nathalie.core@univ-amu.fr (NC); harold.cremer@univ-amu.fr (HC)

**Competing interests:** The authors declare that no competing interests exist.

## Introduction

In the postnatal and adult rodent forebrain, new interneuron precursors are continuously generated by neural stem cell populations along the walls of the lateral ventricles. After their amplification in the subventricular zone (SVZ) and long-distance migration via the rostral migratory stream (RMS) they are added to the preexisting circuitry of the olfactory bulb (OB) (*Alvarez-Buylla and Garcia-Verdugo, 2002*; *Whitman and Greer, 2009*; *Platel et al., 2019*).

OB interneurons produced in this SVZ-RMS-OB neurogenic system show a wide spectrum of phenotypic diversity at the levels of morphology, terminal position, connectivity, and neurotransmitter use (*Whitman and Greer, 2007*). Lineage studies demonstrated that this diversity relies on the existence of defined microdomains of predetermined neural stem cells in the ventricular-subventricular zone [V-SVZ (*Merkle et al., 2007*; *Ventura and Goldman, 2007*; *Lledo et al., 2008*; *Fiorelli et al., 2015*; *Chaker et al., 2016*)].

A key question concerns the molecular mechanisms underlying this diversity. It has been shown that Gli1 activation by Sonic Hedgehog (SHH) is necessary to generate Calbindin-expressing periglomerular neurons (CB-N) in the ventral aspect of the ventricular wall (*Ihrie et al., 2011*). Moreover, the zinc-finger transcription factors (TF) Zic1 and Zic2 act as inducers of Calretinin (CR)-expressing GABAergic interneurons in the dorso-septal region (*Tiveron et al., 2017*). However, the high diversity of the postnatal generated interneuron subtypes suggests that more complex molecular cascades and cross regulatory interactions are put in place to define the stem cell compartment at the necessary resolution.

This is exemplified by the regulation of a group of OB interneurons that, in addition to GABA, use dopamine as their main neurotransmitter (DA-N). This neuron type is generated by neural stem

cells located in the dorsal and dorso-lateral aspects of the ventricular wall. Pax6 is a key determinant of this cell type (*Hack et al., 2005*; *Kohwi et al., 2005*) and is expressed in the entire lineage from stem cells to neurons (*Hack et al., 2005*; *de Chevigny et al., 2012b*). In addition to this positive transcriptional regulation, post-transcriptional mechanisms have been shown to be crucial for the negatively control of DA-N production. Indeed, *Pax6* mRNA expression in the postnatal ventricular wall is not restricted to the DA-N producing dorsal progenitor pool, but extends far into the lateral region where other cell types, including purely GABAergic granule cells and CB-N, are produced. However, the presence of the microRNA miR-7 in a Pax6-opposing ventro-dorsal gradient precludes *Pax6* mRNA translation and restricts protein expression, and consequently DA-N phenotype, to the dorsal region (*de Chevigny et al., 2012a*). Thus, complex molecular events implicating transcriptional and post-transcriptional control mechanisms underlie the functional diversity of OB interneurons.

The Ventral Homeodomain Protein 1 (Vax1), an intracellular mediator of SHH signaling, is expressed in ventral territories of the developing mouse forebrain as well as in ventral aspects of the developing eye (*Hallonet et al., 1999*; *Ohsaki et al., 1999*). In both systems, the expression pattern of *Vax1* is complementary to that of *Pax6*, and genetic studies provided evidence for cross regulatory interactions between both factors (*Hallonet et al., 1998*; *Bertuzzi et al., 1999*; *Hallonet et al., 1999*; *Stoykova et al., 2000*; *Bäumer et al., 2002*; *Mui et al., 2005*).

At embryonic stages, constitutive *Vax1* mutants show a strong decrease in GABAergic interneurons in the developing neocortex, indicating an essential function in their generation (*Taglialatela et al., 2004*). *Vax-1* homozygous mutants die generally at perinatal stages and only few 'escapers' survive for a few weeks after birth. In these animals, the entire postnatal SVZ-RMS-OB neurogenic system is severely compromised, showing accumulation of precursors in the SVZ and severe disorganization of the RMS (*Soria et al., 2004*), altogether precluding a detailed analysis of Vax1-function at later stages.

In an attempt to understand the regulatory cascades underlying postnatal OB interneuron diversity, we identified *Vax1* as a potential candidate. Indeed, based on a high-resolution gene expression screen, comparing the postnatal pallial and subpallial OB lineages, we found that *Vax1* mRNA is present in a ventro-dorsal gradient along the lateral wall of the forebrain ventricles. Functional studies using conditional mutants demonstrate that Vax1 is essential for the generation of CB-N by the ventral stem cell pool. Moreover, we show that Vax1 acts as negative regulator of DA-N OB fate via downregulation of the pro-dopaminergic factor Pax6. Finally, we provide data suggesting that this repressor function is, at least in part, mediated by induction of miR-7.

## Results

### *Vax1* is expressed in a ventro-dorsal gradient along the lateral ventricle

We investigated gene expression during postnatal OB neurogenesis by in vivo electroporation of neural stem cells in the lateral and dorsal aspects of the forebrain lateral ventricle at postnatal day 1 (P1), followed by the isolation of homotypic cohorts at different time points by microdissection and FACS. Microarray analyses provided detailed insight into gene expression changes between the two neurogenic lineages ('in space') and during the progression from stem cells to young neurons ('in time'; *Figure 1A*; for detail see *Tiveron et al., 2017*).

These analyses showed that *Vax1* was confined to the neurogenic lineage derived from the lateral ventricular wall (*Figure 1B*). *Vax1* mRNA was induced at low levels at 1 day post-electroporation (dpe) when most GFP-positive cells were transit amplifying precursors [*Boutin et al., 2008*; *Figure 1B*]. Expression strongly increased at 2dpe and remained stably high at 4dpe, when most cells were migratory neuronal precursors, before steeply decreasing at 7dpe when cells arrived in the OB and emigrated from the RMS to invade the granule cell (GCL) and the glomerular (GL) layers (*Tiveron et al., 2017*). In comparison, isolates from dorsally electroporated brains showed no obvious *Vax1* mRNA expression over all analyzed time points (*Figure 1B*).

Next, we investigated *Vax1* co-expression with known markers of defined neuronal subsets or differentiation stages in the postnatal V-SVZ. In the absence of a Vax1 antibody that provided reliable signals on postnatal tissue sections, we combined in situ hybridization for *Vax1* mRNA with immunohistochemistry for the proteins PAX6, ASCL1, KI67, and DLX2 (*Figure 1C–H*). *Vax1* mRNA always

**Figure 1.** *Vax1* is expressed in the lateral V-SVZ. (**A**) Representation of the strategy used for transcriptomic analysis in time and space in the dorsal and lateral OB lineages. pCX-GFP plasmid was introduced into neural stem cells (NSCs) residing within the dorsal or lateral V-SVZ and GFP-positive cells were isolated by FACS at different time points after electroporation (Elpo). The mRNA content was analyzed by micro-array (*Tiveron et al., 2017*). (**B**) Quantification of *Vax1* mRNA expression detected by micro-array analysis in dorsal (brown) and lateral (purple) progenies during neurogenesis. (**C–H**) In situ hybridization revealing *Vax1* mRNA (in blue) combined with immuno-histochemistry using antibodies detecting (in brown) PAX6 (**C, C', G**), ASCL1 (**D, D'**), DLX2 (**E, E', H**) or KI67 (**F, F'**) proteins in the V-SVZ (**C–F**) or RMS (**G, H**) at postnatal day 3 (**P3**). (**C'–F'**) High magnification of cellular staining in the V-SVZ (area indicated by the yellow bracket in C-F). Arrows (**C'**): examples of strong PAX6 only positive cell in the dorso-lateral SVZ; blue staining underneath labels cells from a distinct plane. Arrow heads (**E', F'**): double positive cells for DLX2 and KI67, respectively. High magnification of the RMS highlights the differential expression of *Vax1* and Pax6 along the dorso-ventral axis (**G',G"**) and the co-localization with Dlx2 (**H',H"**). (**I**) Schematic
*Figure 1 continued on next page*

*Figure 1 continued*

representation of gene expression profile in different cell types of the neurogenic sequence. Circular arrow indicates proliferating cells. LV: lateral ventricle, RG: radial glia, TAP: transit amplified precursor, VZ: ventricular zone, SVZ: sub-ventricular zone. D: dorsal, L: lateral, S: septal, V: ventral. Scale bars: 100 µm (**C–F**), 20 µm (**C'–F'**), 50 µm (**G–H**).

showed a gradient-like distribution along the lateral ventricular wall, with highest expression in the ventral-most aspect and extending far into dorsal regions (*Figure 1C,D,E,F*). Fainter expression was observed along the septal wall. In the lateral wall, *Vax1* mRNA was excluded from the VZ but expressed in the underlying SVZ (*Figure 1C',D',E',F'*). *Vax1* mRNA in the lateral wall was generally non-overlapping with PAX6, a marker for the dorsal stem cell pool and a determinant for the DA-N lineage (Figure C,C'). *Vax1* mRNA-positive cells rarely expressed ASCL1 (3.3%±0.32), a marker for transit amplifying precursors (*Figure 1D,D'*), but the vast majority was labeled with a DLX2 antibody (90%±0.89) (*Figure 1E,E'*; arrowheads), confirming the preferential expression of *Vax1* in migratory neuronal precursors (*Doetsch et al., 2002*; *Figure 1I*). Finally, about one third (32.6%±3.07) of the *Vax1*+ cells in the SVZ expressed the proliferation marker KI67 (*Figure 1F,F'*, arrowheads), likely representing mitotic neuroblasts. In the RMS, *Vax1* mRNA was also present in a ventro-dorsal gradient, complementary to PAX6 immunostaining (*Figure 1G*), although this organization was less evident than in the SVZ, probably due to cell intermingling in this migratory compartment (*Figure 1G', G''*). Like in the SVZ, *Vax1* mRNA in the RMS co-localized strongly with DLX2 immunoreactivity (*Figure 1H*, close up H', H').

Altogether, the combination of microarray studies in defined neuronal lineages and histological approaches led to the conclusion that *Vax1* is expressed in the lateral and ventral SVZ in a subset of proliferating precursors and in most neuroblasts, the latter maintaining expression during their migration in the RMS.

### *Vax1* is necessary for the generation of calbindin-positive interneurons

Previous work demonstrated that CB-N destined for the GL are generated from the ventral-most region of the anterior lateral ventricles (LV), and SHH signaling has been implicated in their specification (*Merkle et al., 2007*; *Ihrie et al., 2011*). As *Vax1* is strongly expressed in this area, and has been shown to act as an intracellular mediator of SHH signaling (*Take-uchi et al., 2003*; *Furimsky and Wallace, 2006*; *Zhao et al., 2010*), we first asked if the TF is implicated in the generation of the CB-N subtype.

*Vax1* conditionally mutant mice (Vax1cKO) (*Hoffmann et al., 2016*) were bred to R26tdTomato mice to monitor CRE-induced recombination and to follow the distribution and fate of mutant and control cells over time. We used postnatal in vivo brain electroporation to express CRE protein in the lateral wall (*Figure 2A*). Since targeting of the *Vax1*-positive ventral region of the ventricular wall with DNA-based expression constructs is inefficient, we used *Cre* mRNA, that is highly efficient for the transfection of stem cells along the entire wall, including the most ventral aspect (*Bugeon et al., 2017*). Animals were electroporated at P0 and analyzed 15 days later, when labeled neurons reached the OB and integrated into the GCL and GL (*Figure 2B*). Quantification of labeled neurons in the GCL and GL revealed no significant differences between control and mutant mice (*Figure 2C*; *Figure 2—figure supplement 2*). TH-positive PGC were also not significantly affected (Figure 4I). However, there was a significant loss in the small population of CB-positive neurons in the GL (*Figure 2D,E*).

Thus, *Vax1* expression in the ventro-lateral-derived neurogenic lineage is necessary for the correct generation of CB-N in the GL.

### *Vax1* regulates *Pax6* during OB neurogenesis

In situ hybridization indicated that *Vax1* mRNA was expressed in a ventro-dorsal gradient (*Figure 1*). To confirm the existence of such a gradient, we micro-dissected V-SVZ tissue from the dorsal, dorso-lateral, and ventro-lateral regions of the ventricular walls of postnatal and adult mice and subjected the isolates to RT-qPCR analyses for *Vax1* mRNA. In agreement with the histological data, *Vax1* mRNA showed a steep ventro-dorsal gradient, opposed to, and partially overlapping with, the well-described localization of *Pax6* mRNA, that extends dorso-ventrally (*Figure 3A*, *Figure 3—figure*

**Figure 2.** *Vax1* is necessary for the production of Calbindin-positive interneurons in the olfactory bulb. (**A**) Representation of the *Vax1* conditional allele (Vax1cKO) and the the inducible reporter *tdTomato* allele in the *Rosa26* locus (*R26tdTom*). Right panel: strategy used to recombine the *Vax1* mutant allele in the V-SVZ cells in the lateral wall at postnatal day 0 (P0). TdTomato (Tom)-positive cells were analyzed 15 days post-electroporation (dpe) in the olfactory bulb (OB). (**B**) Images showing the distribution of Tom+ cells (in red) in the OB at 15dpe in control and mutant brains. Nuclei (in blue) are stained by Hoechst. (**C**) Quantification of granule cells (GC) number in the OB GCL in both conditions. Data are shown as means ± SD, dots represent individual animals. WT: n = 12, Vax1cKO: n = 17. (**D**) Images showing Calbindin+ (in green) and Tom+ cells in the GL at 15dpe. Arrow heads indicate double stained neurons. High magnification of representative double positive cells is shown below. (**E**) Quantification of the percentage of Calbindin+

*Figure 2 continued on next page*

**Figure 2 continued**

neurons among the Tom+ PGC population (WT: n = 11, Vax1cKO: n = 17) showing reduction of CB-N in the mutant. GCL: granule cell layer, GL: glomerular layer. *p≤0.05. Scale bars: 200 µm (**B**), 50 µm (**D**).

The online version of this article includes the following source data and figure supplement(s) for figure 2:

**Source data 1.** Quantification of tdTomato+ granule cells and Calbindin+ PGC in *Vax1* mutant.

**Figure supplement 1.** CRE-recombination of the *Vax1^flox* allele in progenitors induce a substantial reduction of *Vax1* mRNA expression in the homozygote mutant (Vax1cKO) compared to WT animal.

**Figure supplement 2.** Quantification of tdTomato-positive periglomerular cell (PGC) number in the OB of *Vax1* mutant brains.

**Figure supplement 2—source data 1.** Quantification of tdTomato+ PGC in *Vax1* mutant.

---

*supplement 1*; *de Chevigny et al., 2012a*). This observation appeared significant for two main reasons. First, several studies provided evidence that *Vax1* can negatively regulate *Pax6* expression during development (*Bertuzzi et al., 1999*; *Hallonet et al., 1999*; *Mui et al., 2005*). Second, repression of *Pax6* translation along the lateral wall is necessary to confine Pax6 protein, and consequently DA-N phenotype, to the dorsal stem cell pool (*de Chevigny et al., 2012a*).

Based on this information, we hypothesized that *Vax1* is implicated in *Pax6* down-regulation in the postnatal SVZ. To test this, we overexpressed Vax1 in the dorsal and lateral neurogenic lineages by electroporation and investigated the impact on Pax6 expression. A *Vax1* expression plasmid (pCAG-Vax1), or an empty control vector, was co-electroporated with pCX-GFP into either the dorsal or the lateral ventricular wall (*Figure 3B*). Two days later, animals were sacrificed and intensity of Pax6 immunostaining in GFP-positive cells was measured in the dorsal and dorso-lateral SVZ (*Figure 3C,D,E*). GFP-positive cells generated in both compartments showed a significant decrease in Pax6 expression levels at this early time point (*Figure 3D,E*).

Then we asked if Pax6 expression was affected at late time points, after the arrival of newborn neurons in the OB GL. Indeed, 15 and 25 days after pCAG-Vax1 electroporation into the dorsal wall the proportion of GFP-positive neurons that showed Pax6 expression was reduced by over 80% (*Figure 3F,G*).

We conclude that *Vax1* has the capacity to act as a negative regulator of *Pax6* expression during postnatal OB neurogenesis.

### *Vax1* negatively regulates dopaminergic phenotype

DA-N in the OB GL are derived from the dorsal and dorso-lateral aspects of the ventricle walls (*Merkle et al., 2007*; *Fernández et al., 2011*; *de Chevigny et al., 2012a*) and *Pax6* expression is necessary and sufficient for the acquisition of this neurotransmitter phenotype (*Hack et al., 2005*; *Kohwi et al., 2005*). We asked if *Vax1* overexpression in Pax6-positive cells was sufficient to inhibit the generation of DA-N in the OB.

We first targeted the dorsal compartment (*Figure 4A–D*), where the majority of DA-N are generated (*Fernández et al., 2011*; *de Chevigny et al., 2012a*). Ectopic expression of *Vax1* led to a significant loss of TH/GFP-positive neurons 15 days later in the OB (*Figure 4B*). Loss of TH-positive cells was robust over time and could be observed at 25 and 60 dpe (*Figure 4C*). Two observations pointed toward the specific loss of dopaminergic neurons. First, the number of the second major identified neuron type that is generated in the dorsal ventricular wall, CR-N of the glomerular layer (*Fernández et al., 2011*; *Tiveron et al., 2017*), was unaffected by *Vax1* expression (*Figure 4B*), arguing against a fate shift toward this neuron type (*Tiveron et al., 2017*). Second, the density of total GFP+ cells in the GL of *Vax1*-electroporated animals was significantly reduced (*Figure 4D*), whereas the number of GFP+ granule cells was not affected (*Figure 4—figure supplement 1*). As dopaminergic neurons represent a substantial population of all dorsal generated periglomerular cells such an overall loss is coherent with a loss of the DA-N subtype.

Next, we targeted the lateral ventricular wall (*Figure 4E–G*), where smaller but still significant numbers of DA-N are produced from a dorso-lateral stem cell pool (*Figure 4F*). Overexpression of *Vax1* in the lateral wall induced a significant loss of TH-positive neurons in the GL 15 days later (*Figure 4F*). At the same time point, numbers of CB-N and CR-N were unchanged (*Figure 4F*), indicating again that no phenotypic switch toward these subtypes occurred. The density of GFP+ cells

**Figure 3.** *Vax1* inhibits PAX6 expression in the V-SVZ and the OB. (**A**) Quantitative RT-PCR revealing *Vax1* and *Pax6* gene expression in tissue micro-dissected from three distinct areas of the V-SVZ. D: dorsal, DL: dorso-lateral, VL: ventro-lateral. (**B**) Strategy design for the *Vax1* gain-of-function experiment. The *Vax1* expressing plasmid (pCAG-Vax1) was introduced into lateral or dorsal progenitors in combination with pCX-GFP by electroporation at P1. Brains were analyzed at different time points in the V-SVZ or the OB. (**C**) Representative images showing simultaneous expression of PAX6 and GFP proteins in dorsal or lateral lineage in the V-SVZ. (**D**) High-magnification images illustrating the downregulation of Pax6 in GFP+ cells after electroporation of lateral V-SVZ by *Vax1*. White arrows: GFP/Pax6 double positive cells, yellow arrows point to cells with reduced or absentPax6 expression. (**E**) Quantification of PAX6 mean intensity in control or *Vax1*-overexpressing (OE) V-SVZ GFP+ cells from dorsal (D, n = 6 for the control, n = 7 for Vax1 condition) or lateral (L, n = 6 for the CTL, n = 7 for Vax1 condition) walls. (**F**) Images showing simultaneous expression of PAX6 and GFP in the OB glomerular layer of control or Vax1OE brains. Arrow head: double GFP/PAX6-positive cells; yellow arrow: GFP only cells. (**G**) Quantification of GFP+PAX6+ neurons in the OB GL at 15dpe (n = 6 for the CTL, n = 6 for Vax1OE) and 25dpe (n = 8 for the CTL, n = 6 for Vax1OE). PGC: periglomerular cell. *p≤0.05, **p≤0.01. Scale bars: 50 µm (**C,F**), 10 µm (**D**).

The online version of this article includes the following source data and figure supplement(s) for figure 3:

**Source data 1.** Quantification of PAX6 in *Vax1*-overexpressing progenitors and neurons.

**Figure supplement 1.** Quantitative RT-PCR revealing *Vax1* gene expression in adult brain.

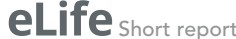

**Figure 4.** Overexpression of *Vax1* in V-SVZ neural stem cells inhibits dopaminergic phenotype. (**A**) Experimental design (left) for the electroporation of NSCs in the dorsal wall with pGAC-Vax1 + pCX-GFP. Images (right) showing expression of Tyrosine Hydroxylase (TH) and Calretinin (CR) in the OB glomerular layer 15 days after electroporation in *Vax1* over-expression (OE) and control brains. White arrow head: GFP/CR- double positive neuron, yellow arrow head: GFP/TH double positive neuron, yellow arrow: GFP-only cell. (**B**) Histogram showing the reduction of the density of GFP+

*Figure 4 continued on next page*

*Figure 4 continued*

periglomerular cells (PGC) in the Vax1OE OB (CTL n = 15, Vax1 n = 14). (**C**) The quantification of TH+ and CR+/GFP-positive cells shows a large decrease of the proportion of dopaminergic neurons among the total GFP+ cells in the OB of Vax1 condition (TH n = 14, CR n = 15) compared to control (n = 13/14). (**D**) The reduction of the TH+ population is sustained with time as it is still observed at 25- (CTL n = 7, Vax1 n = 5) and 60- (CTL n = 5, Vax1 n = 4) days post electroporation. (**E**) Experimental design (left) for the electroporation of NSCs in the lateral wall with pGAC-Vax1 + pCX-GFP. Representative images (right) of immunostaining with TH, Calbindin (CB), and CR antibodies in the OB GL. Arrow head: example of double positive staining with GFP for each marker. (**F**) Histogram presenting the quantification of the three different neuronal populations among the GFP+ neurons in the OB of control (n = 10 for each marker) or Vax1OE (TH n = 11, CB and CR n = 9) conditions. (**G**) Histogram showing the density of GFP + PGC in both conditions (CTL n = 10, Vax1 n = 10). (**H**) Lateral NSCs of Vax1cKO: rosa26tdTom brains were electroporated at birth with pCX-CRE and neuronal phenotype was analyzed in OB at 15 dpe. Representative images of TH staining in the GL of control or *Vax1* deficient OB. Arrow head: GFP+ cells co-labelled with TH. Insert: high magnification of a double positive neuron. (**I**) Histograms presenting the percentage of TH+ neurons among Tom + PGC (CTL: n = 12, three independent litters; Vax1: n = 12, three independent litters). A slight increase of the TH+ population was observed in absence of *Vax1* compared to control but statistical test (Mann Whitney U test) failed to give significant p values (p=0.16). **p≤0.01, ****p≤0.0001. All scale bars: 20 µm except in H (50 µm).

The online version of this article includes the following source data and figure supplement(s) for figure 4:

**Source data 1.** Quantification of OB neuronal subpopulations in *Vax1*- overexpressing or *Vax1* mutant mice.

**Figure supplement 1.** Forced expression of *Vax1* has no effect on cell density in the OB granule cell (GC) layer, 15 days after electroporation of dorsal (CTL n = 15, Vax1 n = 14) or (**B**) lateral (n = 10 for both conditions) V-SVZ progenitors.

was also unaffected in both GL (*Figure 4G*) and GCL (*Figure 4—figure supplement 1*). Thus, *Vax1* overexpression specifically inhibits DA-N phenotype of newborn neurons in the OB.

We also investigated whether *Vax1* loss-of-function had a positive impact on DA-N phenotype in the OB. NSCs along the lateral wall of Vax1cKO animals were electroporated with pCX-CRE (*Figure 4H*) and the proportion of TH-positive neurons in the OB was analyzed 15 days later. These analyses failed to show a significant increase in DA-N at a confidence level of p≤0.05 (*Figure 4I*).

We conclude that *Vax1*, likely via regulation of *Pax6*, has the capacity to negatively control the generation of DA-N for the OB. Moreover, these data show that while *Vax1* is necessary for the generation of CB-N, it is not sufficient.

### *Vax1* induces miR-7 expression in the lateral wall

Previous work demonstrated that mature microRNA miR-7 is expressed in a ventro-dorsal gradient along the lateral ventricular wall and post-transcriptionally inhibits Pax6 protein expression. This interaction confines the generation of DA-N to the very dorso-lateral aspect (*de Chevigny et al., 2012a*). As *Vax1* and miR-7 are expressed in a similar gradient, we hypothesized that the repression of Pax6 protein expression by *Vax1* is mediated by miR-7. To address this idea, we overexpressed Vax1 together with GFP in the lateral stem cell compartment and isolated GFP-positive cells 2 days later by microdissection, dissociation and flow cytometry cell sorting (*Figure 5A*). qRT-PCR analyses demonstrated that augmented *Vax1* expression (*Figure 5B*) led to a strong increase in miR-7 levels (*Figure 5C*), suggesting that *Vax1* regulates *MiR-7* expression. In agreement, bioinformatical analyses of the proximal promoters of the three *MiR-7* loci (*MiR-7–1, MiR-7–2, MiR-7b*) identified a significant match of the Vax1 DNA binding motif within 500 bp of the transcription start site of each of the three *MiR-7* loci present in the mouse genome (*Figure 5D*). Altogether, these results suggest that the negative impact of *Vax1* on Pax6 expression is, at least in part, mediated via the positive regulation of *MiR-7*.

## Discussion

Here, we show that *Vax1* is strongly expressed in the ventral stem cell compartment of the OB neurogenic system where it is necessary for the generation of CB-N. In addition to this actively phenotype-determining function, our data show that *Vax1* acts as a negative regulator of Pax6, likely via the induction of miR-7, thereby restricting the generation of DA-N, which are generated in a neighboring progenitor domain (*Figure 5E*; *de Chevigny et al., 2012a*).

During nervous system development, determination of neuronal phenotype is controlled by the combinatorial expression of transcription factors (*Flames et al., 2007*; *Guillemot, 2007*; *Lai et al., 2016*). Moreover, cross regulatory interactions between such TFs have been shown to tightly define

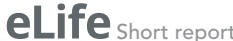

**Figure 5.** *Vax1* induces the expression of miR-7 in the lateral V-SVZ. (**A**) Strategy used to determine the expression of microRNAs in Vax1-overexpressing progenitors. PCAG-Vax1 and pCX-GFP were simultaneously introduced into NSCs by electroporation of the lateral wall of postnatal P1 brains. Lateral V-SVZ was dissected out 2 days after electroporation and GFP+ cells were isolated by flow cytometry (FACS) to perform quantitative RT-PCR analysis. (**B**) Quantification of *Vax1* mRNA level by qRT-PCR in control and Vax1OE conditions, normalized to beta-actin and reported in Vax1 condition as relative level to control, validating the overexpression of *Vax1* after electroporation. (**C**) Quantification of miR-7 expression in both conditions. Expression level of miR-7 was normalized by invariant expression of microRNA let-7a and reported in Vax1 condition as relative level to control. Experiments in B and C were performed in triplicate, and data were obtained from (**B**) two independent biological replications or (**C**) three technical replications. (**D**) Genome browser images representing the chromosomal portions encoding the three *MiR-7* loci (depicted in pink). *Mir-7–1* lies within an intronic sequence of the Hnrnpk gene whereas *MiR-7–2* and *MiR-7b* reside within intergenic sequences. Vax1-binding sites found in the upstream regulatory region of the three *MiR-7* are represented by red boxes. (**E**) Model of cross-regulatory interaction between *Vax1*, miR-7, and *Pax6* in the lateral V-SVZ to control the number of dopaminergic neurons generated by the neural stem cells regionalized in this aspect. This model is supported by our present data and previous work (*de Chevigny et al., 2012a*) where it was shown that miR-7 was required to inhibit PAX6 expression in lateral NSCs to produce the correct number of dopaminergic neurons in the postnatal OB. Here, we propose that *Vax1* acts upstream of miR-7 by positively regulating its expression and consequently inhibiting PAX6. However, it is also possible that *Vax1* directly represses the expression of *Pax6* mRNA (dashed line) by acting on its promoter (*Mui et al., 2005*). Additionally, *Vax1* is required to generate Calbindin neurons from the ventral aspect of the lateral V-SVZ.

The online version of this article includes the following source data for figure 5:

**Source data 1.** Quantification of expression level of *Vax1* and miR-7 in V-SVZ cells.

progenitor domains that generate specific neuron subtypes (*Jessell, 2000*; *Sagner and Briscoe, 2019*). Such spatial information has to be maintained during postnatal and adult stages and neuronal output has to be adapted to the needs of the ongoing neurogenesis in the OB. In agreement, regionalization of the postnatal stem cell compartment has been shown to depend on spatially

restricted and combinatorial expression of TFs like, for example, Pax6, Emx1, Gsx1/2, Gli1/2, or Zic1/2 (*Alvarez-Buylla et al., 2008*; *Weinandy et al., 2011*; *Fiorelli et al., 2015*; *Angelova et al., 2018*).

CB-N are produced by NSCs positioned in the ventral aspect of the V-SVZ (*Merkle et al., 2007*) and SHH signaling, via its effector GLI1, has been implicated in the specification of this subtype (*Ihrie et al., 2011*). Interestingly, previous work demonstrated that Vax1 expression is positively controlled by SHH signaling (*Hallonet et al., 1999*; *Take-uchi et al., 2003*; *Furimsky and Wallace, 2006*). In light of our finding that *Vax1* deletion also leads to specific loss of CB-N, it appears probable that *Vax1* acts downstream of SHH expression in the ventral SVZ to control CB-N production for the OB.

In addition to this local role in CB-N generation, *Vax1* regulates the generation of a neighboring neuron type. Indeed, its expression extends in a gradient far into dorsal regions of the ventricular wall, where Pax6 is expressed and acts as a key component of DA-N generation (*de Chevigny et al., 2012a*). Forced expression of *Vax1* in the *Pax6*-positive domains was sufficient to reduce PAX6 protein expression and to inhibit the production of DA-N, but not of other neuron types, in the OB. This strongly indicates that *Vax1* acts as a repressor of Pax6, comparable to the situation in the developing eye (*Bertuzzi et al., 1999*; *Hallonet et al., 1999*; *Mui et al., 2005*).

Loss-of-function of *Vax1* through targeted electroporation with a CRE expression vector in the lateral wall of conditional mutants did not lead to a statistically significant increase in DA-N at the classically used confidence level of p≤0.05. The observed tendency toward DA-N increase was, however, quite robust over several independent electroporation approaches implicating a large cohort of animals. We decided to include these data, as they complement the gain-of-function approach and as we believe that they could be biologically relevant. Indeed, the cell population in the lateral wall that will be able to induce DA-N fate after deletion of *Vax1* is probably quite small. A large proportion of cells targeted by electroporation in the dorso-lateral wall do not express *Vax1* and these cells will follow their normal differentiation program after recombination. Only cells in intermediate positions, that express sufficiently high Pax6 levels to be able to induce the DA-N phenotype, while at the same time having sufficient *Vax1* levels to suppress this differentiation pathway, will show a DA-N phenotype after *Vax1* removal. In addition, other factors, like lower recombination efficiency of the *Vax1* allele versus the *R26tdTomato* allele, might further diminish the number of cells in which the impact of gene inactivation can be studied (*Long and Rossi, 2009*; *Luo et al., 2020*). Thus, while our electroporation approach allows targeting and manipulating specific stem cell compartments, it also has limitations that restrict interpretation.

Finally, the implication of microRNAs introduces an additional level of complexity into the cross regulatory machinery that underlies OB interneuron fate determination. In previous work, we demonstrated that PAX6 protein, and consequently DA-N production, is confined to the dorsal aspect of the lateral ventricle wall by post-transcriptional regulation of the *Pax6* 3'UTR by the microRNA mir-7. The latter is, like *Vax1*, expressed in a ventro-dorsal and *Pax6* opposing gradient. Thus, it is tempting to speculate that the repression of Pax6 by *Vax1* is, at least in part, indirect and produced via activation of miR-7. Our finding that overexpression of *Vax1* in vivo induced a strong increase in miR-7 levels, and the presence of Vax1-binding sites in all three *MiR-7* promoter regions supports such a scenario. Indeed, regulatory networks implicating transcription factors and miRNAs have been described in other contexts of developmental neurogenesis. For example, in the developing spinal cord, progenitor domains producing defined types of interneurons depend on the cross regulation between the transcription factors Olig2 and Irx3. This interaction is fine-tuned by the induction of miR-17–3 p that represses Olig2 and refines the boundary between domains (*Chen et al., 2011*). Thus, such regulatory modules could be a general mechanism to assure the precise definition of progenitor compartments.

In conclusion, we identified in the postnatal brain a regulatory network, based on transcription factors and miRNAs, which controls the regionalization of the stem cell compartment (*Figure 5E*). The observation that the ventro-dorsally oriented *Vax1* gradient (*Figure 3—figure supplement 1*) as well as the graded expression of *Pax6* and miR-7 (*de Chevigny et al., 2012a*) are maintained in adult stages points to the possibility that this regulatory cascade is also active in adult neurogenesis.

# Materials and methods

**Key resources table**

| Reagent type (species) or resource | Designation | Source or reference | Identifiers | Additional information |
|---|---|---|---|---|
| Strain, strain background (*Mus musculus*) | *Vax1*$^{flox}$ | PMID:27013679 | RRID:MGI:5796178 | |
| Strain, strain background (*Mus musculus*) | *Gt(ROSA)26Sor*$^{tm14}_{(CAG-tdTomato)Hze}$ (Ai14) | Jackson Laboratories | RRID:IMSR_JAX:007914 | |
| Strain, strain background (*Mus musculus*) | CD1 | Charles River | Crl : CD1(ICR) RRID:IMSR_CRL:022 | |
| Sequence-based reagent | Mouse *Vax1* cDNA | GenBank | BC111818 | |
| Sequence-based reagent | *Cre* mRNA | Miltenyi Biotec | 130-101-113 | |
| Recombinant DNA reagent | pCX-EGFP (plasmid) | PMID:17934458 | | Dr Xavier Morin (CNRS, Aix-Marseille University) |
| Recombinant DNA reagent | pCX-CRE (plasmid) | PMID:17934458 | | Dr Xavier Morin (CNRS, Aix-Marseille University) |
| Antibody | Anti-digoxigenin (Sheep polyclonal) | Roche | 11093274910 RRID:AB_514497 | IHC (1:1000) |
| Antibody | Anti-Pax6 (Rabbit polyclonal) | Millipore | AB2237 RRID:AB_1587367 | IF, IHC (1: 1000) |
| Antibody | Anti-Ascl1 (mouse monoclonal) | BD Biosciences | 556604 RRID:AB_396479 | IHC (1:100) |
| Antibody | Anti-Ki67 (mouse monoclonal) | BD Biosciences | 550609 RRID:AB_393778 | IHC (1:200) |
| Antibody | Anti-Dlx2 (guinea pig polyclonal) | Prof. K. Yoshikawa, Osaka University, Osaka, Japan | PMID:16707790 | IHC (1: 2000) |
| Antibody | Anti-Calbindin D-28K (Rabbit polyclonal) | Millipore | AB1778 RRID:AB_2068336 | IF (1 :1000) |
| Antibody | Anti-Calbindin D-28K (mouse monoclonal) | Swant | 300 RRID:AB_10000347 | IF (1:3000) |
| Antibody | Anti- Calretinin (mouse monoclonal) | Synaptic systems | 214111 RRID:AB_2619906 | IF (1:1000) |
| Antibody | Anti-Tyrosine hydroxylase (chicken polyclonal) | Avès Labs | TYH, RRID:AB_10013440 | IF (1:1000) |
| Commercial assay or kit | SYBR GreenER qPCR SuperMix | ThermoFisher Scientific | 11762100 | |
| Commercial assay or kit | miScript SYBR Green PCR Kit | Qiagen | 218073 | |
| Commercial assay or kit | miRCURY LNA miRNA PCR Assay | Qiagen | 339306 | |
| Commercial assay or kit | miRNAeasy kit | Qiagen | 217004 | |
| Software, algorithm | ZEN Blue | Zeiss | RRID:SCR_013672 | |
| Software, algorithm | Axiovision imaging system | Zeiss | RRID:SCR_002677 | |
| Software, algorithm | Fiji | http://fiji.sc | RRID:SCR_002285 | |
| Software, algorithm | ImageJ | https://imagej.nih.gov/ij/ | RRID:SCR_003070 | |

*Continued on next page*

*Continued*

| Reagent type (species) or resource | Designation | Source or reference | Identifiers | Additional information |
|---|---|---|---|---|
| Software, algorithm | R Commander | https://CRAN.R-project.org/package=Rcmdr | RRID:SCR_001905 | |
| Software, algorithm | FlowJo | https://www.flowjo.com/solutions/flowjo | RRID:SCR_008520 | |

## Animals

All animal procedures were carried out in accordance to the European Communities Council Directie 2010/63/EU and approved by French ethical committees (Comité d'Ethique pour l'expérimentation animale no. 14; permission numbers: 00967.03; 2017112111116881 v2). Animals were held on a 12 h day/night cycle and had access to food and water ad libitum. Animals of both sexes were used for experiments. CD1 mice (Charles River, Lyon, France) were used for in vivo electroporation and expression pattern analyses. $Vax1^{flox}$ (Vax1cKO) conditional mutants (Hoffman 2016) and $Rosa26^{tdTomato}$ reporter mice (Ai14, Jackson Laboratories, USA, RRID:IMSR_JAX:007914) were bred on a mixed C57BL/6*CD1 genetic background. *Vax1flox* genotyping was performed with *Vax1flox* forward: 5'-GCCGGAACCGAAGTTCCTA; *Vax1wt* forward: 5'-CCAGTAAGAGCCCCTTTGGG, reverse 5'-CGGATAGACCCCTTGGCATC. Ai14 genotyping was performed with *R26wt* forward: 5'-AAGGGAGCTGCAGTGGAGTA, reverse 5'-CCGAAAATCTGTGGGAAGTC; *R26tdTom* forward: 5'- CTGTTCCTGTACGGCATGG and reverse: 5'- GGCATTAAAGCAGCGTATCC.

## Plasmid and in vivo electroporation

The full-length rat cDNA sequence of *Vax1* was excised from pCMV2-Rn-Vax1-FLAG (a gift of Kapil Bharti) and subcloned into pCAGGS vector to produce pCAG-Vax1. Postnatal day 0 (P0) or day 1 (P1) pups were electroporated as previously described (*Boutin et al., 2008*; *de Chevigny et al., 2012a*) with plasmid DNA or RNA (*Bugeon et al., 2017*). CRE Recombinase mRNA (130-101-113, a generous gift from S. Wild and A. Bosio, Miltenyi Biotec, Bergisch Gladbach, Germany) and pCX-CRE (*Morin et al., 2007*) were used at a concentration of 0.5 µg/µl. Recombination efficiency was tested by qRT-PCR from tdTomato+ cells isolated by FACS (*Figure 2—figure supplement 1*). In CD1 pups, pCX-EGFP (*Morin et al., 2007*) was co-injected with pCAG-Vax1 or empty pCAGGS (as control) in a 1:2 molecular ratio to label the electroporated cells. Targeting of the dorsal or lateral wall of the lateral ventricle was directed by distinct orientation of the electrodes. Brains were collected at different time points after electroporation.

## In situ hybridization and immunohistochemistry

For all procedures, tissues were fixed by intracardiac perfusion with ice-cold 4% paraformaldehyde (wt/vol) in phosphate buffered saline (PBS) and cryoprotected in 30% sucrose solution in PBS. Brains were sliced coronally using either a microtome (Microm Microtech, France) or a cryostat (Leica Biosystems, France).

Mouse *Vax1* cDNA clone (GeneBank: BC111818, Imagene) was used to produce the anti-sense RNA probe. Combined in situ hybridization (ISH) and immunohistochemistry (IHC) were performed as previously described (*Tiveron et al., 1996*) on 16 µm cryosections at postnatal day 3 (P3) using anti-digoxigenin antibody (Sheep polyclonal, 1:1000, Roche, 11093274910, RRID:AB_514497). Two days exposure to alkaline phosphatase (AP) substrate NBT/BCIP (Promega, S3771) was necessary to detect AP-DIG-labeled *Vax1* mRNA at maximal level. Subsequently, antibodies directed against PAX6 (rabbit polyclonal, 1: 1000, Millipore, AB2237, RRID:AB_1587367), KI67 (mouse IgG1k clone B56, 1:200, BD Biosciences, 550609, RRID:AB_393778), ASCL1 (mouse, clone 24B72D11.1, 1:100, BD Biosciences, 556604, RRID:AB_396479), or DLX2 (rabbit polyclonal, 1: 2000 a gift from Prof. K. Yoshikawa) were applied to the ISH treated sections and were revealed with secondary antibodies coupled to horseradish peroxidase (Jackson ImmunoResearch Laboratories, UK), using 3, 3'-Diaminobenzidine (Invitrogen, 750118) as a substrate. Sections were finally mounted in fluoromount medium and analyzed by light microscopy using Axiovision Rel. 4.8 software (Zeiss, Germany).

For immunofluorescence, 50 µm floating sections were blocked in PBS supplemented with 0.5% Triton-X100, 10% foetal calf serum (FCS) and incubated overnight at 4°C in PBS, 0.1% Triton, 5%

FCS with primary antibodies: anti-Calbindin D-28K (rabbit polyclonal, 1:1000, Millipore, AB1778, RRID:AB_2068336 or mouse IgG1, 1:3000, Swant, 300, RRID:AB_10000347), anti-Calretinin (mouse IgG1, clone 37C9, 1:1000, Synaptic systems, 214111, RRID:AB_2619906), anti-Tyrosine hydroxylase (chicken IgY, 1:1000, Avès labs, TYH, RRID:AB_10013440). After washing in PBS, Alexa Fluor-conjugated secondary antibodies (Jackson ImmunoResearch Laboratories) were applied diluted at 1:500 in blocking solution for 2 hr at room temperature. After staining of cell nuclei with Hoechst 33258 (Invitrogen, H3569), sections were mounted with Mowiol (Sigma-Aldrich, 81381).

## RNA isolation, qRT-PCR, cell dissociation, and FACS

Animals (P1-P3) were decapitated and brains were cut into 300 µm thick sections using a Vibrating-Blade Microtome (Thermo Scientific, HM 650V). V-SVZ and RMS tissues were micro-dissected under a binocular microscope and kept in cold Hank's balanced salt solution (HBSS, Gibco, 14170120). RNA was extracted using the miRNAeasy kit (Qiagen, 217004) or by Trizol reagent (life technologies, 15596026) according to manufacturer instructions, allowing the recovery of long and short RNAs. For *Pax6* and *Vax1* expression analysis, cDNA was prepared using superscript III reverse transcriptase (ThermoFisher Scientific, 12574–030) following manufacturer instructions and quantitative PCR was performed on a BioRad CFX system using SYBR GreenER qPCR SuperMix (ThermoFisher Scientific, 11762100) in technical triplicates. ß-Actin was used as reference gene. Primers used for mRNA detection are the following: Beta Actin forward 5'-CTAAGGCCAACCGTGAAAAG and reverse 5'-ACCAGAG GCATACAGGGACA; Pax6 forward 5'-TGAAGCGGAAGCTGCAAAGAAA and reverse 5'-TTTGGCCCTTCGATTAGAAAACC; Vax1 forward 5'- GCTTCGGAAGATTGTAACAAAAG and reverse 5'- GGATAGACCCCTTGGCATC.

For miRNA expression, cDNA was prepared using miScript II (Qiagen, 218160) and qPCR was performed using miScript SYBR Green PCR Kit (Qiagen, 218073) together with miRCURY LNA miRNA PCR Assay (Qiagen, 339306) on a StepOne Real-Time PCR System (Applied Biosystems). Let-7a was used as normalizer miRNA. Three independent technical replications for each condition were processed in triplicates.

To generate single-cell suspensions for FACS, dissected tissues were subjected to cell dissociation using a Papain solution and mechanical dissociation as described previously (*Lugert et al., 2010*). Cells were resuspended in HBSS/Mg/Ca supplemented with 10 mM HEPES (Gibco, 15630080), 40 µg/ml DNase I (Roche, 10104159001), 4.5 g/L Glucose (Gibco, A2494001) and 2 mM EDTA, filtered through a 30 µm pre-separation filter (Miltenyi Biotec, 130-041-407) and sorted on MoFlo Astrios EQ cytometer (Beckman- Coulter) gating on GFP- or tdTomato- positive population.

## Bioinformatics analysis

Significant Vax1 DNA binding motif (JASPAR Core position-weight matrix MA0722.1) occurrences in the three *MiR-7* loci (*MiR7-1, MiR7-2, MiR7b*) proximal promoters (defined as 500 bp upstream of the transcription start site) from the *Mus musculus* mm10 genome were determined using FIMO from the MEME suite (*Bailey et al., 2009*) with a maximum *p*-value limit of 0.001. FIMO-derived *p*-values of motif matches in the promoters were 0.000173 (*MiR7-1*), 0.000333 (*MiR7-2*), and 0.000806 (*MiR7b*). Genome browser images were created using the UCSC Genome Browser.

## Image analyses

All images were analyzed blind to the experimental condition. Optical images were acquired with an Axioplan2 ApoTome microscope (Zeiss, Germany) using ZEN software (Zeiss, RRID:SCR_013672), and processing was performed using Fiji software (RRID:SCR_002285, [*Schindelin et al., 2012*]) or ImageJ (NIH, https://imagej.nih. gov/ij/, RRID:SCR_003070). ISH/IHC double staining experiments were analyzed by counting labelled SVZ cells along the lateral ventricular wall on 40X magnification images (n = 3 SVZ for Ki67 and Dlx2, n = 8 SVZ for Ascl1). For cell counting in V-SVZ and OB tissues from electroporation experiments, a minimum of three sections per brain (n = animal) were analyzed and the totality of GFP+ or Tomato+ cells of each section were counted. To measure mean intensity of fluorescence after antibody staining, ROI were applied on individual electroporated cells.

## Statistical analysis

Control and *Vax1* electroporation assays were performed from the same litter and were reproduced in two or three independent experiments for lateral or dorsal electroporation, respectively. Histograms were drawn with GraphPad Prism version 8 (GraphPad Software, San Diego, CA). Data are presented as mean ± SD. Each sample (n = animal/brain) is represented on histograms by dot. Statistical analyses were performed using R software (RRID:SCR_001905) and R Commander Package (https://CRAN.R-project.org/package=Rcmdr). The non-parametric two-tailed Mann Whitney U test was performed for all in vivo experiments, on pooled experimental repetitions when appropriate. Differences were considered statistically significant when p≤0.05.

## Acknowledgements

We thank the members of the Cremer lab for support and critical reading of the manuscript. We are grateful to S Wild and A Bosio from Miltenyi Biotec for providing CRE-mRNA. We thank Kapil Bharti (NIH, Bethesda, USA) for sharing Vax1 cDNA. We thank the local PiCSL-FBI core facility (IBDM, AMU-Marseille) supported by the French National Research Agency through the « Investments for the Future' program (France-BioImaging, ANR-10-INBS-04) as well as the IBDM animal facility. We are grateful to AMUTICYT Cytometry and Cell Sorting Core facility, AMU, UMR-S 1076. This work was supported by the Agence National pour la Recherche (grants ANR- 13-BSV4-0013 and ANR- 17-CE16-0025), Fondation pour la Recherche Medicale (FRM) 'Label Equipe FRM' and Fondation de France (FDF) grant FDF70959 to HC. AE was supported for a postdoctoral fellowship from the Swiss National Funds. NIH grants to PLM in support of this work are P50 HD12303, R01 HD072754, R01 HD082567, P30 CA23100, P30 DK063491, and P42 ES010337. HMH was supported by NIH K99 HD084759.

## Additional information

### Funding

| Funder | Grant reference number | Author |
| --- | --- | --- |
| Agence Nationale de la Recherche | 13-BSV4-0013 | Nathalie Coré<br>Christophe Béclin<br>Harold Cremer |
| Agence Nationale de la Recherche | 17-CE16-0025 | Nathalie Coré<br>Christophe Béclin<br>Harold Cremer |
| Fondation de France | FDF70959 | Nathalie Coré<br>Christophe Béclin<br>Harold Cremer |
| Fondation pour la Recherche Médicale | EQU201903007806 | Nathalie Coré<br>Christophe Béclin<br>Harold Cremer |
| National Institutes of Health | P50 HD12303 | Pamela L Mellon |
| National Institutes of Health | R01 HD072754 | Pamela L Mellon |
| National Institutes of Health | R01 HD082567 | Pamela L Mellon |
| National Institutes of Health | P30 CA23100 | Pamela L Mellon |
| National Institutes of Health | P30 DK063491 | Pamela L Mellon |
| National Institutes of Health | P42 ES010337 | Pamela L Mellon |
| National Institutes of Health | K99 HD084759 | Hanne M Hoffmann |
| Swiss National Science Foundation | P2BSP3_175013 | Andrea Erni |

The funders had no role in study design, data collection and interpretation, or the decision to submit the work for publication.

## Author contributions
Nathalie Coré, Conceptualization, Formal analysis, Validation, Investigation, Visualization, Writing - original draft, Writing - review and editing; Andrea Erni, Andrew J Saurin, Investigation; Hanne M Hoffmann, Pamela L Mellon, Resources; Christophe Beclin, Supervision, Funding acquisition, Investigation, Writing - original draft, Writing - review and editing; Harold Cremer, Supervision, Funding acquisition, Writing - original draft, Writing - review and editing

## Author ORCIDs
Nathalie Coré (iD) https://orcid.org/0000-0003-3865-4539
Harold Cremer (iD) https://orcid.org/0000-0002-8673-5176

## Ethics
Animal experimentation: All animal procedures were carried out in accordance to the European Communities Council Directie 2010/63/EU and approved by French ethical committees (Comité d'Ethique pour l'expérimentation animale no. 14; permission numbers: 00967.03; 2017112111116881v2).

## Decision letter and Author response
Decision letter https://doi.org/10.7554/eLife.58215.sa1
Author response https://doi.org/10.7554/eLife.58215.sa2

# Additional files

## Supplementary files
• Transparent reporting form

## Data availability
All data generated or analysed during this study are included in the manuscript.

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
