## [Decision Letter]

**Acceptance summary:**

The olfactory bulb is a rare brain region that incorporates newly born neurons in the adult. The molecular pathways that give rise to the diversity of olfactory bulb neuron types during adult neurogenesis are poorly understood. This study reveals a key role for the transcription factor *Vax1* in guiding cell fate decisions of adult-born neurons, promoting formation of calbindin interneurons and suppressing formation of dopaminergic neurons.

**Decision letter after peer review:**

Thank you for submitting your article "Regulatory interactions between *Vax1*, *Pax6* and *miR-7* regionalize the lateral SVZ during mice olfactory bulb neurogenesis" for consideration by *eLife*. Your article has been reviewed by three peer reviewers, and the evaluation has been overseen by a Reviewing Editor and Gary Westbrook as the Senior Editor. The reviewers have opted to remain anonymous. The reviewers have discussed the reviews with one another and the Reviewing Editor has drafted this decision to help you prepare a revised submission.

As the editors have judged that your manuscript is of interest, but that additional experiments are required before it is published, we would like to draw your attention to changes in our revision policy that we have made in response to COVID-19 (https://elifesciences.org/articles/57162). First, because many researchers have temporarily lost access to the labs, we will give authors as much time as they need to submit revised manuscripts. We are also offering, if you choose, to post the manuscript to bioRxiv (if it is not already there) along with this decision letter and a formal designation that the manuscript is "in revision at *eLife*". Please let us know if you would like to pursue this option. (If your work is more suitable for medRxiv, you will need to post the preprint yourself, as the mechanisms for us to do so are still in development.)

Summary:

Neurons in the olfactory bulb are replenished throughout life, but mechanisms underlying the maintenance of neuronal diversity remain poorly defined. In this study, the authors identify a key role for *Vax1* and *Pax6* in the cell fate decisions that control differentiation of particular olfactory bulb interneurons.

Essential revisions:

– Data implicating a functional role for *Mir-7* are preliminary; it is advised that claims be bolstered or toned down.

– Comments from all reviewers are provided below in case they are helpful. Reviewer #3 in particular requested additional data and information prior to rendering a final decision.

Reviewer #1:

Core et al. study the transcription factor *Vax1* in the context of cell fate using the SVZ-RMS-OB adult neurogenesis model. The novel part of the data seems incremental and while suggestive it's not always compelling; Particularly so the claims of specificity. Some analyses and controls could be improved.

1) Figure 1 – The combination of ISH and IC is legitimate but the data are not compelling. The overlap in Figure 1G-H are good examples. This is also evident in the qualitative description of overlap. In any event, these data are not very clean. The paper would benefit from a quantitative approach. For example, a double-labeling smFISH expression analysis would be more successful.

2) Figure 2 – The comparison of GCL tdt+ numbers (Figure 2 B,C ) should be compared to PGL numbers. The current comparison to 2D,E is not a fair comparison (apples to apples…). For claim a specificity effect, they should compare to other cell types in the PGL in the same animals.

3) Figure 3D – The large scale gradient is convincing but not very novel. To drive this conclusion home, a similar analysis that is done in 3A should be conducted at a single cell level.

In C-F the claim for specificity is weak. They should show that expression of other TFs are not hampered in the GFP cells. It’s possible that the mere over expression of a *Vax1* impacts other proteins in a negative way. I am not from the field but it seems to me that a better control is needed (e.g. at the very least a scrambled version of Vax1OE).

4) Figure 4 – in 4B the CR distribution seems to be different – it has a double peak as compared to controls. More appropriate statistics may show this difference.

4I distributions are very similar. The text description of these as being marginally significant is misleading.

5) The *miR-7* data are robust but remain anecdotal and currently not very natural to add it to the storyline of the paper that remains intact without it.

Reviewer #2:

The manuscript by Core et al. examines the molecular players that display spatial segregation and regulate neonatal progenitor fate from the SVZ to the olfactory bulb. They build on previous work from their lab and others that have reported specific function of *PAx6* and *mir-7* in neuron fate determination. The manuscript is well-written and uses an array of approaches to address their questions. Overall, the data are strong and well presented. My enthusiasm is to some extent limited as outlined in the following comments:

1) Data regarding the contribution of *miR-7* are limited. I think more data are either necessary to show that *Vax1* works through *mir-7* to regulate neuronal fate or the authors should change their writing and conclusion.

For example, in the Abstract: "We provide evidence that this repression occurs via activation of microRNA *miR-7*, targeting *Pax6* mRNA." I think it is overselling the data presented in this manuscript. Much more data would be required for this conclusion. For example, the *mir-7* data could be left out of the Abstract but presented at the end of the Introduction as well as discussed.

2) Figure 1: please show analysis of the in situ/immune data.

3) Better images for Figure 3C would be helpful to actually see the decrease in *Pax6* intensity.

4) The sentence: "However, we stably observed a tendency for an increase over independent electroporation experiments (3 experiments for each condition with a total of 12 animals/ condition, Figure 4I)." should be removed. There is no effect.

Reviewer #3:

During the process of brain development in mammals, neuronal fates are determined by the combinatorial expression of transcription factors. In the olfactory bulb (OB), different subsets of interneurons are constantly supplied from the subventricular zone (SVZ) not only in neonates but also during adulthood. These OB interneurons demonstrate a wide variety of phenotypes in morphology, migration, connectivity, and the use of neurotransmitters. It has been shown that this diversity of interneurons is generated depending on the stem-cell microdomains along the walls of lateral ventricles. In this study, the authors tried to clarify the molecular mechanism underlying the positional identity of neural stem-cells by analyzing gene expression during postnatal OB neurogenesis.

Using the conditional knockout of a homeodomain protein, Vax1, as well as the in vivo electroporation of the Vax1-expression vector, the authors demonstrated new findings that provide fundamental insights into the cross-regulatory interactions that determine neuronal phenotypes in the mammalian brain. Major findings in this study are: 1) Vax1 is a key determinant for the generation of Calbindin-positive interneurons in the ventral compartment of the SVZ; 2) Vax1 negatively regulates the transcription of Pax6 whose expression is essential for the generation of dopaminergic neurons in the dorsal compartment; and 3) Repression of Pax6 occurs by microRNA *miR-7* targeting the Pax6 mRNA.

These findings are novel and will provide new insights into our understanding of fate determination of developing neurons. The paper is clearly written and easy to understand. Experiments in this study are elegantly designed. I therefore support publication of this manuscript in *eLife* if the reviewer's comments are properly responded.

Specific comments:

1) In the loss-of-function experiment in Figure 2, it would be helpful if the authors could show how efficiently Vax1 expression is suppressed in the targeted tdTomato positive cells by using in situ hybridization.

2) Also in the gain-of-function experiment in Figure 3, the levels of Vax1 derived from the electroporated vector DNA should be shown for the GFP-positive cells, by in situ hybridization.

3) In this manuscript, the authors analyzed regulatory interactions among Vax1, Pax6, and *miR-7* only in neonates. Since interneurons are constantly regenerated during adulthood, it would be helpful, if the authors could demonstrate the data showing that the same regulatory cross-talk can be seen in the adult neurogenesis of interneurons.

4) What is the mechanism of Vax1-mediated *miR-7* enhancement? Do they directly interact with each other? Is there any binding site for Vax1 within the promoter region of the *miR-7* gene?

5) Cross-regulatory interactions for *Vax1*, *Pax6*, and *miR-7* are nicely shown in this study. It would be helpful if the authors could discuss this regulation strategy in the context of neurogenesis during brain development in order for the paper to attract a wider research audience other than those investigating transcription factors.

---

## [Author Response]

Reviewer #1:1) Figure 1 – The combination of ISH and IC is legitimate but the data are not compelling. The overlap in Figure 1G-H are good examples. This is also evident in the qualitative description of overlap. In any event, these data are not very clean. The paper would benefit from a quantitative approach. For example, a double-labeling smFISH expression analysis would be more successful.

We quantified the percentage of double stained cells for the different markers used in Figure 1 D-F. Moreover, in the RMS (Figure 1 G,H) cells migrate partially intermingled, making a graded distribution of *Vax1* and *Pax6* positive cells, as is obvious in the SVZ, less evident. We added higher magnification images to illustrate the differential expression of *Vax1* in the ventral (Figure 1G’) and dorsal (Figure 1G’’) aspects of the RMS. We also added higher magnifications to illustrate the co-expression of DLX2 and *Vax1* in the RMS (Figure 1H’ and H’’). The Results section was changed accordingly.

2) Figure 2 – The comparison of GCL tdt+ numbers (Figure 2 B,C ) should be compared to PGL numbers. The current comparison to 2D,E is not a fair comparison (apples to apples…). For claim a specificity effect, they should compare to other cell types in the PGL in the same animals.

New data showing the PGC quantification was added to the manuscript as Figure 2—figure supplement 1B. Please also note that, in addition to the CB+ PGC population now presented in Figure 2E, we already analyzed in the same animals the number of TH^+^ PGC (Figure 4I).

3) Figure 3D – The large scale gradient is convincing but not very novel. To drive this conclusion home, a similar analysis that is done in 3A should be conducted at a single cell level.In C-F the claim for specificity is weak. They should show that expression of other TFs are not hampered in the GFP cells. It’s possible that the mere over expression of a Vax1 impacts other proteins in a negative way. I am not from the field but it seems to me that a better control is needed (e.g. at the very least a scrambled version of Vax1OE).

Here, the reviewer’s comments/questions are in part a bit unclear to us and we hope we got the main points.

To our knowledge the existence of large-scale gradients in the postnatal SVZ is still a relatively new concept. Designing and performing a single cell analysis experiment that provides reliable information about such gradients is not easy and, to our eyes, exceeds the scope of this study. Use of a scrambled protein as control for gain-of-function experiments in vivo is a rather unusual approach in the field. As far as we know, such a control has so far not been applied. Finally, specific *Pax6*/*Vax1* regulatory interactions have been shown in other brain developmental contexts, like for example the eye, arguing for the specificity of the observed effects here.

4) Figure 4 – in 4B the CR distribution seems to be different – it has a double peak as compared to controls. More appropriate statistics may show this difference.4I distributions are very similar. The text description of these as being marginally significant is misleading.

Concerning 4b: We agree that there is some heterogeneity in this specific experiment. However, all samples have been treated exactly the same way and analyzed by the appropriate statistical tests. We do not see which variables can be tested to address the reviewers point.

Concerning Figure 4I: We accept the reviewer’s critique and removed any potentially misleading conclusion concerning these data from the Results section.

5) The miR-7 data are robust but remain anecdotal and currently not very natural to add it to the storyline of the paper that remains intact without it.

We added additional data, reinforcing the link between *Vax1* and *miR-7*, by showing the all three *miR-7* promoters contain putative *Vax1* binding sites. We opted to maintain the data in the manuscript, but tempered our conclusions concerning this aspect. We also removed the mention of *miR-7* from the manuscript title.

Reviewer #2:1) Data regarding the contribution of miR-7 are limited. I think more data are either necessary to show that Vax1 works through mir-7 to regulate neuronal fate or the authors should change their writing and conclusion.For example, in the Abstract: "We provide evidence that this repression occurs via activation of microRNA miR-7, targeting Pax6 mRNA." I think it is overselling the data presented in this manuscript. Much more data would be required for this conclusion. For example, the mir-7 data could be left out of the Abstract but presented at the end of the Introduction as well as discussed.

We accept the reviewer’s critique and react threefold. First, we provide new data showing that all three *miR-7* promoters contain putative *Vax1* binding sites. Second, we tempered our conclusions concerning the *Vax1*-*miR-7* link in the Abstract and the main manuscript. Third, we removed the term *miR-7* from the title.

2) Figure 1: please show analysis of the in situ/immune data.

As requested, we included quantifications of the in situ/immune staining’s. The data are now presented in the Results section.

3) Better images for Figure 3C would be helpful to actually see the decrease in Pax6 intensity.

We added new images illustrating the downregulation of *Pax6* protein after *Vax1* electroporation. The new data are now presented as Figure 3D.

4) The sentence: "However, we stably observed a tendency for an increase over independent electroporation experiments (3 experiments for each condition with a total of 12 animals/ condition, Figure 4I)." should be removed. There is no effect.

We agree and removed the phrase from the Results section of manuscript.

Reviewer #3:1) In the loss-of-function experiment in Figure 2, it would be helpful if the authors could show how efficiently Vax1 expression is suppressed in the targeted tdTomato positive cells　by using in situ hybridization.

To address the efficiency of CRE mediated recombination we expressed the recombinase by in vivo electroporation in control and *Vax1*/Ai14 double mutants and isolated transfected tdTomato-positive cells by FACS. The quantitative RT-PCR results, now shown in Figure 2—figure supplement 1, clearly show a massive reduction in *Vax1* mRNA levels after CRE expression, demonstrating efficient recombination.

This approach was necessary as in light of the relatively small number of transfected cells after in vivo electroporation – and facing the cellular complexity of the postnatal SVZ- in situ hybridization is not suited to give a quantitative notion of gene-knockout.

2) Also in the gain-of-function experiment in Figure 3, the levels of Vax1 derived from the electroporated vector DNA should be shown for the GFP-positive cells, by in situ hybridization.

As for the knockout, we combined electroporation, here of an overexpression plasmid, with FACS and qRT-PCR. We found the expected large increase in *Vax1* mRNA levels in the GFP positive fraction. This control is shown in Figure 5B.

3) In this manuscript, the authors analyzed regulatory interactions among Vax1, Pax6, and miR-7 only in neonates. Since interneurons are constantly regenerated during adulthood, it would be helpful, if the authors could demonstrate the data showing that the same regulatory cross-talk can be seen in the adult neurogenesis of interneurons.

In previous work we demonstrated that the dorsoventral *Pax6* and the ventrodorsally oriented *miR-7* gradients are maintained in adult stages (de Chevigny et al., 2012b, Supplemental Figures 1 and 4). Here, we show now, as the new

Figure 3—figure supplement 1, that the postnatal ventrodorsally oriented *Vax1* gradient is also maintained in the adult, allowing to speculate that the mechanism remains active. This new data on adult *Vax1* expression is now mentioned and discussed in the manuscript.

It would of course be tempting to test the validity of the *Vax1*/*Pax6*/*miR-7* interactions in adults. However, in vivo electroporation in adults is far less efficient than in the postnatal, leading only to low amounts of transfected cells in the OB. As our analyses depend on precise quantification of large numbers of cells in gain- and loss of-function experiments, meaningful data cannot be expected from such approaches.

Viral transduction, that is normally used in adults, cannot be targeted to specific subregions of the SVZ. It appears conceivable to design complex transgenic approaches to target specific SVZ compartments with CRE, but such experiments, if possible, would represent a long-term project on its own that exceeds what can be done here. We hope this is acceptable.

4) What is the mechanism of Vax1-mediated miR-7 enhancement? Do they directly interact with each other? Is there any binding site for Vax1 within the promoter region of the miR-7 gene?

Following the reviewer’s suggestion, we investigated the presence of *Vax1* binding sites in the *miR-7* regulatory regions. Indeed, we found such sites in all three *miR-7* promoters, in agreement with a potential direct regulation. This new data are now presented in Figure 5D.

5) Cross-regulatory interactions for Vax1, Pax6, and miR-7 are nicely shown in this study. It would be helpful if the authors could discuss this regulation strategy in the context of neurogenesis during brain development in order for the paper to attract a wider research audience other than those investigating transcription factors.

To place our results in a wider context, we discuss now the OLIG2/Irx3/miR-17-3p regulatory network, that is used to define progenitor domains in the developing spinal cord.